Classification of the drifting data streams using heterogeneous diversified dynamic class-weighted ensemble

http://orcid.org/0000-0003-3019-8364 Sarnovsky Martin martin.sarnovsky@tuke.sk
Kolarik Michal
Department of Cybernetics and Artificial Intelligence, Faculty of Electrical Engineering and Informatics, Technical University in Kosice , Kosice , Slovakia
Cano Alberto
Electronic publication date: 2021 Apr 1
Publication date: 2021
Volume: 7
Electronic Location ID: e459
Received 2020 Nov 12; Accepted 2021 Mar 5
Copyright: © 2021 Sarnovsky and Kolarik
Copyright year: 2021
Copyright holder: Sarnovsky and Kolarik
License: This is an open access article distributed under the terms of the Creative Commons Attribution License, which permits unrestricted use, distribution, reproduction and adaptation in any medium and for any purpose provided that it is properly attributed. For attribution, the original author(s), title, publication source (PeerJ Computer Science) and either DOI or URL of the article must be cited.
License URL: https://creativecommons.org/licenses/by/4.0/

Keywords: Ensemble learning, Concept drift, Data streams, Adaptive ensemble

Funding: Slovak Research and Development Agency APVV-16-0213 The work was supported by the Slovak Research and Development Agency under the contract No. APVV-16-0213 Knowledge-based approaches for intelligent analysis. The funders had no role in study design, data collection and analysis, decision to publish, or preparation of the manuscript.

==============================
Data streams can be defined as the continuous stream of data coming from different sources and in different forms. Streams are often very dynamic, and its underlying structure usually changes over time, which may result to a phenomenon called concept drift. When solving predictive problems using the streaming data, traditional machine learning models trained on historical data may become invalid when such changes occur. Adaptive models equipped with mechanisms to reflect the changes in the data proved to be suitable to handle drifting streams. Adaptive ensemble models represent a popular group of these methods used in classification of drifting data streams. In this paper, we present the heterogeneous adaptive ensemble model for the data streams classification, which utilizes the dynamic class weighting scheme and a mechanism to maintain the diversity of the ensemble members. Our main objective was to design a model consisting of a heterogeneous group of base learners (Naive Bayes, k-NN, Decision trees), with adaptive mechanism which besides the performance of the members also takes into an account the diversity of the ensemble. The model was experimentally evaluated on both real-world and synthetic datasets. We compared the presented model with other existing adaptive ensemble methods, both from the perspective of predictive performance and computational resource requirements.

Introduction

Nowadays, the size of data is growing in a much faster fashion than in the past. Information is being collected from household appliances, tools, mobile devices, vehicles, sensors, websites, social networks, and many other devices. An increasingly large number of organizations are starting to analyze large volumes of data, as the information obtained from these data can provide a competitive advantage over other businesses. Data collection from devices is often continuous, and data come in the form of data streams.

Data stream classification is an active field of research, as more data sources can be considered as streaming data. When solving classification tasks using streaming data, the data generation process is not strictly stationary, and its underlying structure may change over time. The changes in the underlying data distribution within the streams may result in dynamic, non-stationary target concepts (Gama et al., 2014; Žliobaite, 2010). This phenomenon is called concept drift, and from the perspective of the training of classification models on the drifting data, the most crucial requirement is the ability of the model to adapt and incorporate new data into the model in order to react to potential changes (Barddal et al., 2017). In concept drift, the adaptive learning algorithms are advanced machine learning methods that can reflect the changing concepts in data streams in real time. Multiple approaches were proposed to extend the standard machine learning models with the ability to adapt to the changes in streams, including drift detectors (Gonçalves et al., 2014; Baena-García et al., 2006) and various sliding window techniques (Bifet & Gavaldà, 2007).

Ensemble models are a popular classification method, often providing better performance when compared to the standard machine learning models (Breiman, 1996, 2001; Freund & Schapire, 1996). When processing dynamic data streams, where the concepts change over time, dynamic adaptive ensembles present a suitable method that retains long-present historical concepts and covers newly appearing ones. Ensemble methods proved to be capable of handling streaming data by updating its base learners (Kolter & Maloof, 2007; Bifet et al., 2015; Brzeziński & Stefanowski, 2011; Gomes et al., 2017) in either block-based, batch mode or instance-based, incremental mode. One of the crucial aspects of the ensemble classifiers for static data is to ensure the diversity of the base classifiers within the ensemble. In contrast to the diversity of the ensembles for the static data (Carney & Cunningham, 2000; Kuncheva & Whitaker, 2003), which is fairly well studied in the literature, there are fewer studies dealing with the ensemble’s diversity in the presence of concept drift (Brzezinski & Stefanowski, 2016; Abassi, 2019).

The main objective of the work presented in this paper is to present the design and implementation of a novel adaptive ensemble classification algorithm. The primary motivation of this study is to design a model capable of handling various types of drifts. The proposed method can be characterized as a heterogeneous, chunk-based approach that utilizes different types of base classifiers within the ensemble. The model also uses Q statistic as a metric to measure the diversity between the ensemble members and dynamic weighting scheme. We assume that the creation of a heterogeneous ensemble consisting of different base learners with an adaptation mechanism that ensures the diversity of its members can lead to a robust ensemble that is able to handle the drifting data efficiently. To confirm these assumptions, we conducted an extensive experimental study, where we evaluated the proposed model performance on the selection of both real-world and synthetic, generated datasets with different types of concept drift. We also compared the proposed method to 12 adaptive ensemble methods, including state-of-the-art adaptive ensembles. The main objective was to compare the performance of the methods using standard evaluation metrics, as well as training times and resource consumption.

The paper is organized as follows. The background section provides basic definitions of the terms related to the data streams and the concept drift therein. The following section is dedicated to the state of the art in the area of the adaptive ensemble models used to handle the concept drift and defines the motivation for the presented approach. The following section describes the designed and implemented adaptive ensemble method. We then describe the datasets used in the experimental evaluation. The experimental results section summarizes the conducted experiments and achieved results. Concluding remarks and possibilities for future work in this area are then summarized.

Background

A data stream is defined as an ordered sequence of data items, which appear over time. Data streams are usually unbounded and ordered sequences of the data elements ⟨x1,x2,…,xj,…,⟩ sequentially appearing from the stream source item by item (Gama, Aguilar-Ruiz & Klinkenberg, 2008). Each stream element is generated using a probability distribution Pj. The time interval between the particular elements on the stream may vary. Particular data elements in the stream are usually of standard size, and most of the data streams are generated at high speed, e.g., the elements in the data streams appear rapidly.

Compared with static data, which is usually analyzed offline, data streams need to be analyzed in real time. The differences in the processing of the data streams compared to the processing of the static data can be summarized according to Gama (2010):The data elements in the stream arrive online or in batches. Elements appearing online are processed one by one, and batches usually have the same size and are processed at once.

The processing system has no control over the order in which data elements appear, either within a data stream or across multiple data streams.

Data streams are potentially unbound in size. The size of the particular elements is usually small. In contrast, the entire data size may be huge, and it is generally impossible to store the whole stream in memory.

Once an element from a data stream has been processed, it is usually discarded. It cannot be retrieved unless it is explicitly stored in memory.

The data streams can be divided into two groups:stationary data streams—the data distribution does not change over time, e.g., the stream elements are generated from a fixed probability distribution;

non-stationary data streams—data are evolving, and the data distribution may change over time. Usually, these changes may also affect the target concepts (classes).

Concept drift

When solving predictive data analytical tasks on the static data, the data distribution usually does not change, and data used for the training and testing of the model have the same distribution. When processing the data streams, we often observe the changing nature of the data. In predictive data stream analytical tasks, we experience a phenomenon called concept drift.

Concept drift is related to the data distribution Pt(x,y), where x=(x1,x2…xn) is a data sample represented by an n-dimensional feature vector appearing at time t, and y represents the target class. The concepts in the data are stable (or stationary) if all the data samples are generated with the same distribution. If there is an x in the interval between t and t + Δ, which holds the expression Pt(x,y) ≠ Pt + Δ(x,y), then concept drift is present (there is a change in the underlying data distribution) (Žliobaite, 2010). Concept drift usually occurs in a non-stationary and dynamically changing environment, where the data distribution or relation between the input data and the target variable changes over time.

The concept drift phenomenon may occur in various real-world data and corresponding applications (Žliobaite, Pechenizkiy & Gama, 2016):computer systems or networks, through network intrusion detection, where new techniques and methods may appear (Liu et al., 2017; Mukkavilli & Shetty, 2012);

industry, when dynamic data streams are produced by sensors in production equipment and machines (Lin et al., 2019; Zenisek, Holzinger & Affenzeller, 2019);

marketing and management, when users change their buying behavior and their preferences (Black & Hickey, 2003; Chiang, Wang & Chu, 2013; Lo et al., 2018);

medical data, e.g., in the case of antibiotic resistance (Stiglic & Kokol, 2011; Tsymbal et al., 2006);

social networks, when users change their behavior and generated content (Lifna & Vijayalakshmi, 2015; Li et al., 2016);

spam categorization, where spam keywords can change over time (Delany et al., 2005; Ruano-Ordás, Fdez-Riverola & Méndez, 2018).

The authors in Tsymbal (2004), Žliobaite (2010) and Khamassi et al. (2019) describe a complex taxonomy of existing drift types. In general, there are several different types of concept drift, based on how the phenomenon occurs within the data stream:Sudden/Abrupt—In this case, the concept change occurs suddenly. A concept (e.g., a target class) is suddenly replaced by another one. For example, in the topic modeling domain, the main topic of interest may unexpectedly switch to a different one.

Incremental—Changes in the data distribution are slower and proceed over time. Changes are not as visible as in a sudden drift, but they gradually emerge. The changes are usually relatively slow and can be observed when comparing the data over more extended time periods.

Gradual—In this drift type, both concepts are present, but over time, one of them decreases, while the other one increases. For example, such a change may reflect the evolution of points of interest, e.g., when a point of interest is gradually being replaced by a newer one.

Re-Occurring—A previously active concept reappears after some time. Re-occurrence may appear in cycles or not (e.g., reappearing fashion trends).

Besides the mentioned drift types, some publications (Gama et al., 2014) distinguish between two kinds of concept drift: real drift and virtual drift. Virtual concept drift is defined by the changes in data distribution but does not affect the target concept. Real concept drift (also called concept shift) represents a change in the target concept, which may modify the decision boundaries.

Figure 1 visualizes the drift types. In concept drift detection, it is also necessary to distinguish between the drift and outlier occurrence. Outliers may produce false alarms when detecting the concept drift.

Figure 1 Concept drift types according to Gama et al. (2014).

(A) Sudden/Abrupt, (B) incremental, (C) gradual, (D) re-occuring.

When processing the non-stationary drifting streams, the necessary feature of the predictive algorithms is their ability to adapt. Some of the algorithms are naturally incremental (e.g., Naive Bayes), while others require significant changes in the algorithm structure to enable incremental processing. Therefore, the learning algorithms applied on the drifting streams are usually extended with a set of mechanisms, which enhance the models with the ability of continuously forgetting the obsolete learned concepts and updating the model with the newly arrived data in the stream. There are several types of models used to handle concept drift. To detect the concept drift in data streams, we can use drift detectors. These can detect possible concept drift by analyzing the incoming data or by monitoring the classifier performance. Detectors process the signal from the data about changes in data stream distribution. Drift detectors usually signalize drift occurrence and trigger the updating/replacement of the classifier. There are several drift detection methods available (Gonçalves et al., 2014), and the Drift Detection Method (DDM) (Gama et al., 2004), the Early Drift Detection Method (EDDM) (Baena-García et al., 2006), and ADWIN (Bifet & Gavaldà, 2007) are the most popular.

For predictive data modeling applied on the drifting streams, advanced adaptive supervised machine learning methods are used. Supervised learning methods used for drifting stream classification could be categorized from several perspectives, depending on how they approach the adaptation (Ditzler et al., 2015; Krawczyk et al., 2017):Active/Passive—Active methods usually utilize drift detection methods to detect the drift and to trigger the model update. Passive methods periodically update the model, without any knowledge of the drift occurrence.

Chunk-Based/Online—Chunk-based methods process the streaming data in batches (each batch consists of a specified fixed number of stream elements). Online methods process the stream elements separately when they appear.

Ensemble models represent a popular solution for the classification of drifting data streams. An ensemble classification model is composed of a collection of classifiers (also called base learners, ensemble members, or experts) whose individual decisions are combined (most often by voting) to classify the new samples (Sagi & Rokach, 2018). The main idea of the ensemble model is based on the assumption that a set of classifiers together can achieve better performance than individual classifiers (Kuncheva & Whitaker, 2003). The selection of the ensemble experts is a crucial factor, as the ideal ensemble consists of a set of diverse base learners. Ensemble models are also suitable for data stream classification, where target concepts change over time. The following section summarizes the use of ensembles in the classification of drifting streams.

Related work

There are several approaches to the design of adaptive ensemble models. Some of them use the same technique as approaches to static data processing, such as Online Boosting (Wang & Pineau, 2013), which is based on the classic Boosting method, extended with online processing capabilities. For adaptation to concept drift, it uses concept drift detection. If a concept drift occurs, the entire model is discarded and replaced by a new model. Another well-known model is the OzaBagging (Oza, 2005) ensemble. Unlike Bagging for static data, OzaBagging does not use random sampling from the training data, but each of the samples is trained k times, which leads to a Poisson distribution.

Further studies have focused on the design of the ensemble models that would be simple (in terms of their run time) and able to adapt to the concept drift dynamically. For example, the Accuracy Weighted Ensemble (AWE) (Brzeziński & Stefanowski, 2011) uses the assignment of weight to the base classifiers based on a prediction error. Old and weak members are gradually being replaced by the new ones, with a lower error rate. The update mechanism is based on the assumption that the latest training chunk will better represent the current test chunk. Another model, Dynamic Weighted Majority (DWM) (Kolter & Maloof, 2007), dynamically changes the weights of the base classifiers in the case of incorrect classification. A new classifier is added if the model incorrectly classifies the training example, and old classifiers are discarded if their weights fall below a threshold value. Online Bagging and Boosting algorithms were recently used as a basis for more advanced streaming ensembles, such as Adaptive Ensemble Size (AES) (Olorunnimbe, Viktor & Paquet, 2018), which dynamically adapts the ensemble size, or an approach (Junior & Nicoletti, 2019), where boosting is applied to the new batches of data and maintains the ensemble by adding the base learners according to the ensemble accuracy rate. Learn++ (inspired by AdaBoost) is an incremental learning ensemble approach consisting of base learners trained on a subset of training data and able to learn the new classes (Polikar et al., 2001). Several modifications of this approach exist, focused on improvement of the number of generated ensemble members (Muhlbaier, Topalis & Polikar, 2004). The Random Forests method is probably the most popular ensemble method on static data at present. Its adaptive version for stream classification, Adaptive Random Forests (ARF), was introduced in Gomes et al. (2017) and has shown a high learning performance on streaming data. More recently, multiple adaptive versions of popular ensemble methods gaining improved performance or achieving speedup in execution have been introduced, e.g., the adaptive eXtreme Gradient Boosting method (Montiel et al., 2020), the streaming Active Deep Forest method (Luong, Nguyen & Liew, 2020), or Random Forests with an implemented resource-aware elastic swap mechanism (Marrón et al., 2019).

All ensemble models work with the assumption of the diversity of the individual classifiers in the ensemble, while the diversity is achieved in different ways. Diversity can help in evolving data streams, as the most suitable method may also change as a result of the stream evolution Pesaranghader, Viktor & Paquet (2018). Diverse ensembles by themselves cannot guarantee faster recovery from drifts, but can help to reduce the initial increase in error caused by a drift Minku, White & Yao (2010). There are several ways to achieve diversity in the ensemble. Either the classifiers are trained on different data samples, or the model is composed of a set of heterogeneous classifiers. Recently, Khamassi et al. (2019) studied the influence of diversity techniques (block-based, weighting-data, and filtering-data) on adaptive ensemble models and designed a new ensemble approach that combines the three diversity techniques. The authors in Sidhu & Bhatia (2018) experimented with a diversified, dynamic weighted majority voting approach consisting of two ensembles (with low and high diversity, achieved by replacing the Poisson (1) with Poisson (κ) distribution in online bagging (Oza & Russell, 2001)). The Kappa Updated Ensemble (KUE) Cano & Krawczyk (2020) trains its base learners using different subsets of features and updates them with new instances with a given probability following a Poisson distribution. Such an approach results in a higher ensemble diversity and outperforms most of the current adaptive ensembles. However, there are not many studies where the model uses the model diversity score as a criterion for the base classifiers in the ensemble (Krawczyk et al. (2017)) as opposed to static data processing, where such a complex model exists (Lysiak, Kurzynski & Woloszynski, 2014). According to Yang (2011), diversity correlates with model accuracy. A suitable diversity metric used in ensembles is a paired diversity Q statistic (Kuncheva, 2006), which provides information about differences between two base classifiers in the ensemble.

Another aspect of the ensemble classifiers is the composition of the base classifiers in the model. The most common are homogeneous ensemble methods, which use the same algorithm to train the ensemble members (Fernandez-Aleman et al., 2019). On the other hand, heterogeneous approaches are based on the utilization of multiple algorithms to generate ensemble members. Such an approach could lead to the creation of more diverse ensembles. For the data stream classification, a Heterogeneous Ensemble with Feature drift for Data Streams (HEFT-Stream) Nguyen et al. (2012) builds a heterogeneous ensemble composed of different online classifiers (e.g., Online Naive Bayes). Adaptive modifications of the heterogeneous ensembles were also successfully applied on the drifting data streams (Van Rijn et al., 2016; Frías-Blanco et al., 2016; Van Rijn et al., 2018; Idrees et al., 2020), and many of them proved suitable to address issues such as class imbalance (Large, Lines & Bagnall, 2017; Fernández et al., 2018; Ren et al., 2018; Wang, Minku & Yao, 2018; Ghaderi Zefrehi & Altınçay, 2020). The approach described in this paper aims to combine the construction of the adaptive heterogeneous ensemble with a diversity-based update of the ensemble members. This approach could result in a robust model, with the adaptation mechanism ensuring that newly added members are as diverse as possible during the ensemble updates. Maintaining the diversity of the overall model can also lead to a reduction of model updates and therefore faster execution during the run time.

Ddcw ensemble method

In the following section, we introduce the design of the Diversified Dynamic Class Weighted (DDCW) ensemble model. The design of the model is based on the assumption that a robust model consists of a collection of heterogeneous base classifiers that are very diverse. When applied to static data, the diversity is used within the ensemble models to tune the combination rule for voting and the aggregation of component classifier predictions. We propose the design of a heterogeneous ensemble model, which combines the dynamic weighting of the ensemble members with the mutual diversity score criterion. The diversity measures are used to rank the members within the ensemble and update their weights according to the diversity value, so the model prefers experts with higher mutual diversity, thereby creating a more robust ensemble. When ranking the base classifiers, the diversity measurement is combined with the lifetime of individual base classifiers in the model. The criterion is expected to cause the importance of the long-lasting base classifiers to gradually fade, which should ensure the relevance of the whole ensemble to evolving and changing data streams.

The model is composed of m ensemble members e1,…, em, trained using each chunk of incoming samples in the stream, as depicted in Fig. 2. Each of those experts e1,,…, em, have assigned weights for each target class. The weights are tuned after each period (after each chunk is processed) based on individual base classifier performance. First, for each class that a base classifier predicts correctly, the weight is increased. Second, after each chunk is processed, the model calculates Q pairwise diversity between each of the ensemble members and uses this value to modify model weights.

Figure 2 Overall scheme of the proposed ensemble model.

Pairwise Q diversity metric is calculated as follows (Kuncheva & Whitaker, 2003): let Z=z1,...,zN be a labeled data set, zj∈Rn coming from the classification problem. The output of a classifier Di is an N-dimensional binary vector yi=[y1,i,...,yN,i]T, such that yj,i=1, if Di recognizes correctly zj, and 0 otherwise, i=1,...,L. Q statistic for two classifiers, Di and Dk, is then computed as: Qi,k=N11N00−N01N10N11N00+N01N10 where Nab is the number of elements zj of Z for which yj,i=a and yj,k=b. Q varies between −1 and 1; classifiers that tend to recognize the same samples correctly will have positive values of Q, and those that commit errors on different objects will render Q negative. For statistically independent classifiers, the value of Qi,k is 0.

The value for each member div_ei in DDCW model is obtained as the average of contributions of individual pair diversities and is calculated as follows: div_ei=1m−1∑k=1,k≠imQi,k. Then, after each period, the lifetime coefficient Ti of each ensemble member is increased. Afterwards, the weights of each of the base classifiers are modified using the lifetime coefficient. After this step, the weights are normalized for each class, and a score is calculated for each target class by the classifier predictions. In the last step, the global ensemble model prediction is selected as a target class with the highest score.

Model weights can be represented as a matrix Wm×c, where m is a number of classifiers in the ensemble, and c is a number of target classes in the data. The weights wi,j directly determine the weight given to classifier i for class j as seen in Table 1. In the beginning, the weights are initialized equally, based on the number of classes in the data. During the process, the individual weights for each base classifier and corresponding target class are modified using the parameter β. The weight matrix allows the calculation of the score of the base classifiers, as well as the score of predicted target classes. The score of the classifier is calculated as ∑j=1cwi,j for each classifier. This score allows the identification of poorly performing base classifiers in the ensemble. However, the score of the target class is calculated as the contribution of weight wi,j of classifier i and its predicted class j.

Table 1 Weight matrix.

	C1	C2	…	Cc	Classifier score	
e1	w1,1	w1,2	…	w1,c	∑j=1cw1,j	
e2	w2,1	w2,2	…	w2,c	∑j=1cw2,j	
…	…	…	…	…	…	
em	wm,1	wm,2	…	wm,c	∑j=1cwm,j	

The weight matrix enables the efficient contribution of each of the ensemble members in building the model. We proceeded from the assumption of having a diverse ensemble. Such a model could consist of members, which perform differently in specific target class values, e.g., some of the members are more precise when predicting a particular class than others and vice versa. In that case, we can use a set of classifiers; each of them focused on a different target class (while the distribution of these classes may change over time). The class weighting alone may lead to an ensemble consisting of similar, well-performing models. But the combination of class weighting with diversity measure can lead to a set of balanced members, more focused on specific classes and complementing each other.

The proposed ensemble model belongs to the category of passive chunk-based adaptive models. In the presented approach, the size of the model dynamically changes according to the global model performance. The minimum size of the model is set by the parameter k, and the maximum size is set by the parameter l.

Each base classifier in the ensemble is assigned with a weight vector for each target class. If a new target class appears in the chunk used in training, the new target will be added to the weight vector for each base classifier. Initially, the ensemble consists of a randomly initialized set from a list of defined base algorithms (Hoeffding Tree or Naive Bayes). Other experts can be added in the following periods (interval representing the chunk of arriving data, where base learners and their weights are modified) until the minimum size of the model is reached, i.e., either the model size is smaller than the defined minimum size or the particular member weight falls under the defined threshold.

In each iteration, the experts are used to predict the target class of incoming samples in the processed chunk (Lines 3–5). If a prediction of an expert is correct, the weights of the particular expert and target class are multiplied by a coefficient β (Line 7). In the case period p has occurred, Q statistic diversity is calculated for each pair of experts in the ensemble, and the weights of each expert is modified using the particular expert’s diversity (Line 12 and 16). This mechanism enables the construction of more robust ensembles, by preferring the diverse base models. The weights of base classifiers are also reduced by the exponential value of their lifetime in the ensemble (Line 15). In this case, the lifetime of the expert represents the number of periods since its addition to the ensemble. The exponential function is used, so the experts are influenced minimally during their initial periods in the ensemble but become more significant for the long-lasting members. This implementation works as a gradual forgetting mechanism of the ensemble model, as the weakest experts are gradually removed from the model and replaced by the new ones.

After the update, the weights are normalized for each target class (Line 24). Afterwards, if the maximum size of the model is reached and the global prediction is incorrect, the weakest expert is removed from the ensemble (Line 27). A new random expert can then be added to the ensemble (Lines 30–31). In each period, all experts where the sum of weights is lower than defined threshold θ are removed from the ensemble. In the end, each sample is weighted by a random uniform value m times, where m represents the actual size of ensemble (Line 41). Each expert is than trained with a new set of incoming samples, with individual weights from the last chunk of data (Line 42). After training, the global predictions of actual samples are retrieved, and the algorithm then continues back to Line 3.

Algorithm 1 Diversified dynamic class weighted ensemble.

Procedure: DDCW({X, Y}, p, k, l, α, β, θ)	
Input: Data and labels {x, y}, chunk size p, min_experts k, max_experts l, fading factor α, multiplier β, threshold θ	
Output: Global predictions G	
1: Experts ← create_random_experts(k);	
2: initialize class weights wi,j	
3: for s = 0,…,n do	
4: for i = 1,…, num_experts(Experts) do	
5: Local_predictions = classify(Expertsi, xs);	
6: if Local_predictions = ys then	
7: wi,L = β * wi,L; ← Multiply weight of particular expert and target class from local prediction by β	
8: end if	
9: end for	
10: if all samples in a chunk are processed then	
11: Local_predictions = classify(Experts, x_s);	
12: Diversity = calculate_diversity(Local_predictions, y_s);	
13: for i = 1,…, num_experts(Experts) do	
14: expert_lifetime ← Increase expert lifetime in each period;	
15: wi = wi – (exp(α * expert_lifetime) – 1)/10;	
16: wi = wi (1 – Diversityi);	
17: end for	
18: end if	
19: for j = 0,…,Class_labels do	
20: Global_predictionsj ← sum(wj);	
21: end for	
22: Global_predictions ← argmax(Global_predictionsj);	
23: if all samples in chunk are processed then	
24: w ← normalize_weights(w);	
25: if Global_predictionss! = ys then	
26: if num_experts(Experts) == l then	
27: {Experts, w, expert_lifetime} ← Remove weakest expert ei based on experts score	
28: end if	
29: if num_experts(Experts) < l then	
30: Expertsnew ← create_random_expert();	
31: wnew ← 1/num_experts(Experts);	
32: end if	
33: end if	
34: {Experts, w, expert_lifetime} ← Remove experts which score is below threshold θ	
35: if num_experts(Experts) < k then	
36: Expertsnew ← create_random_expert();	
37: wnew ← 1/num_experts(Experts);	
38: end if	
39: end if	
40: for i = 1,…, num_experts(Experts) do	
41: Sample_weightss ← random_uniform_weight();	
42: Expertsi ← learn_expert(Expertsi, xs, ys, Sample_weightss);	
43: end for	
44: return Global_predictions;	
45: end for	

Datasets description

To experimentally evaluate the performance of the presented approach, we decided to use both real-world and synthetically generated datasets. We tried to include multiple drifting datasets that contain different types of concept drift. Datasets used in the experiments are summarized in Table 2.

Table 2 Datasets used in the experiments.

(Dataset type) R: real, S: synthetic. (Drift type) A: abrupt, G: gradual, –: none, ?: unknown.

Dataset	Drift type	Dataset type	Samples	Features	Classes	
ELEC	?	R	45,312	6	2	
KDD99	?	R	494,021	41	23	
AIRL	?	R	539,383	7	2	
COVT	?	R	581,012	54	7	
SHUTTLE	?	R	58,000	9	7	
POWERSUPPLY	?	R	29,928	3	24	
CONNECT-4	?	R	65,557	43	3	
BNG_BRIDGES	?	R	1,000,000	13	6	
BNG_BRIDGES1vsAll	?	R	1,000,000	13	6	
BNG_HEPATITIS	?	R	1,000,000	20	2	
BNG_ZOO	?	R	1,000,000	17	7	
BNG_LYMPH	?	R	1,000,000	19	4	
AGRa	A	S	1,000,000	9	2	
AGRg	G	S	1,000,000	9	2	
SEAa	A	S	1,000,000	3	2	
SEAg	G	S	1,000,000	3	2	
STAGGER	A	S	100,000	3	2	
LED	G	S	100,000	24	10	
MIXED_BALANCED	A	S	1,000,000	5	2	
MIXED_IMBALANCED	A	S	1,000,000	5	2	
RBF	–	S	1,000,000	50	4	
RBF_DRIFT	G	S	1,000,000	50	4	
WAVEFORM	–	S	1,000,000	40	3	
WAVEFORM_DRIFT	G	S	1,000,000	40	3	

Real datasets

In our study, we used 12 real datasets, including the frequently used ELEC dataset (Harries, 1999), the KDD 99 challenge dataset (Tavallaee et al., 2009), Covtype (Blackard, 1998), the Airlines dataset introduced by Ikonomovska (http://kt.ijs.si/elena_ikonomovska/data.html), and data streams from the OpenML platform Bischl et al. (2019) generated from a real-world dataset using a Bayesian Network Generator (BNG) Van Rijn et al. (2014). We included a wide range of datasets to evaluate the performance of the DDCW model on datasets with both binary and multi-class targets or with balanced and imbalanced classes, especially some of them, such as KDD99 and Shuttle are heavily imbalanced. To examine the imbalance degree of the datasets, we included the class ratios in the documentation on the GitHub repository (https://github.com/Miso-K/DDCW/blob/master/classratios.txt). As it is difficult to determine the type of a real drift contained in such data, we tried to estimate and visualize possible drift occurrences. Multiple techniques for concept drift visualization exist (Pratt & Tschapek, 2003). We used visualization based on feature importance (using the Gini impurity) and the respective changes within the datasets, as they may signalize changes in concepts in the data. Based on the work described in Cassidy & Deviney (2015), we used feature importance scores derived from the Online Random Forest model trained on the datasets. Such an approach can help to visualize the so-called feature drift, which occurs when certain features stop (or start) being relevant to the learning task. Fig. 3 shows such visualizations for the real-world datasets used in the experiments. The visualization depicts how the importance of the features changes over time in the data. x-axis represents number of samples, y-axis feature indices and the size of the dots correspond to a feature importance in the given chunk.

Figure 3 Feature importance progress in the real-world datasets.

(A) ELEC, (B) airlines, (C) KDD 99, (D) covtype, (E) shuttle, (F) connect-4 (x axis) number of samples (y axis) feature indices, the size of the dots correspond to a feature importance in the given chunk.

Synthetic datasets

Besides the real-world datasets, we used synthetic data streams containing generators with various types of drifts. In most cases (except LED and STAGGER data), we used streams of 1,000,000 samples, with three simulated drifts. We used the Agrawal generator (Agrawal, Imieliński & Swami, 1993) and SEA (Nick Street & Kim, 2001) generators with abrupt and gradual drifts, RBF and Waveform streams without any drift and with simulated gradual drift, a Stagger Concepts Generator (Schlimmer & Granger, 1986) with abrupt drift, an LED (Gordon et al., 1984) stream with gradual drift, and a Mixed stream with an abrupt drift with balanced and imbalanced target attributes.

Experimental results

The purpose of the experiments was to determine the performance of the DDCW model in a series of different tests. We used the python implementation of all models, and the experiments were performed using the Scikit-multiflow framework (Montiel et al., 2018). All experiments were performed on a virtual server equipped with 6 CPU cores and 8 GB RAM.

During the first series of experiments, we aimed to examine the impact of the different setting of the chunk-size parameter in which the model is updated and the diversity is calculated. The model was tested with varying sizes of the chunk, with values set to 100, 200, 300, 400, 500, and 1,000 samples on all considered datasets. The primary motivation of the experiments was to find the most suitable chunk size parameter for different datasets, which will be used to compare the DDCW with other ensemble models. To evaluate the models, we used prequential evaluation (or interleaved test-then-train evaluation), in which the testing is performed on the new data before they are used to train the model.

In the second set of the experiments, the main goal was to compare the performance of the DDCW model with the selected other streaming ensemble-based classifiers. We considered multiple ensemble models: DWM, AWE, Online Boosting, OzaBagging, and the currently best performing streaming ensembles such as ARF and KUE. To analyze the performance of these models, standard classification metrics were used (accuracy, precision, recall, and F1). Besides the comparison of the model performance, we measured the metrics related to resource consumption, such as total time required for training, scoring time of an instance, and memory requirements of the models. During these experiments, we set the chunk size to 1,000 samples (although we included DDCW with a chunk size of 100 samples for comparison) and set the model’s hyper-parameters to create similar-sized ensemble models (min. 5 and max. 20 members in the ensemble).

Performance with the different chunk sizes

In this experiment, we explored the influence of the chunk window on the classifier’s performance on different datasets. The main goal was to find the optimal chunk window size for a particular dataset. We set different chunk sizes and measured the model performance using selected metrics in defined periods (e.g., every 100 samples). We computed the average model performance on the entire dataset using the above-mentioned classification metrics. A comparison of the DDCW classifier accuracy and F1 with different sizes of the chunks is shown in Table 3.

Table 3 Performance of the DDCW model with different chunk sizes.

Bold values show the best results.

Chunk size	100	200	300	400	500	1,000	
	Accuracy	Accuracy	Accuracy	Accuracy	Accuracy	Accuracy	
ELEC	0.849	0.842	0.831	0.847	0.832	0.810	
KDD99	0.995	0.995	0.995	0.996	0.995	0.991	
AIRL	0.636	0.641	0.644	0.645	0.645	0.649	
COVT	0.849	0.842	0.831	0.847	0.832	0.810	
SHUTTLE	0.953	0.955	0.965	0.975	0.953	0.992	
POWERSUPLY	0.156	0.155	0.156	0.153	0.155	0.157	
CONNECT4	0.686	0.693	0.687	0.669	0.674	0.705	
BNG_BRIDGES	0.689	0.703	0.711	0.715	0.718	0.725	
BNG_BRIDGES1vsAll	0.962	0.963	0.964	0.965	0.965	0.966	
BNG_HEPATITIS	0.858	0.867	0.870	0.873	0.874	0.897	
BNG_ZOO	0.876	0.892	0.900	0.905	0.908	0.921	
BNG_LYMPH	0.796	0.811	0.819	0.824	0.830	0.879	
AGRa	0.857	0.871	0.874	0.876	0.878	0.879	
AGRg	0.827	0.847	0.852	0.856	0.859	0.874	
SEAa	0.873	0.876	0.877	0.878	0.881	0.888	
SEAg	0.868	0.874	0.873	0.874	0.877	0.884	
STAGGER	0.946	0.946	0.933	0.938	0.912	0.923	
LED	0.892	0.888	0.882	0.884	0.884	0.860	
MIXED_BALANCED	0.927	0.934	0.935	0.939	0.943	0.964	
MIXED_IMBALANCED	0.924	0.930	0.932	0.936	0.939	0.964	
RBF	0.855	0.872	0.877	0.879	0.881	0.882	
RBF_DRIFT	0.546	0.562	0.573	0.585	0.592	0.601	
WAVEFORM	0.819	0.826	0.830	0.832	0.835	0.837	
WAVEFORM_DRIFT	0.820	0.826	0.829	0.834	0.835	0.836	
	F1	F1	F1	F1	F1	F1	
ELEC	0.815	0.807	0.791	0.811	0.791	0.760	
KDD99	0.570	0.591	0.602	0.581	0.582	0.596	
AIRL	0.531	0.538	0.533	0.537	0.529	0.535	
COVT	0.815	0.807	0.791	0.811	0.791	0.760	
SHUTTLE	0.506	0.510	0.536	0.606	0.672	0.702	
POWERSUPLY	0.109	0.110	0.109	0.105	0.108	0.110	
CONNECT4	0.483	0.486	0.470	0.466	0.481	0.464	
BNG_BRIDGES	0.588	0.609	0.620	0.627	0.632	0.635	
BNG_BRIDGES1vsAll	0.866	0.872	0.873	0.877	0.877	0.883	
BNG_HEPATITIS	0.907	0.913	0.915	0.918	0.918	0.935	
BNG_ZOO	0.784	0.806	0.818	0.825	0.831	0.853	
BNG_LYMPH	0.548	0.577	0.595	0.607	0.618	0.752	
AGRa	0.842	0.858	0.864	0.868	0.868	0.869	
AGRg	0.808	0.830	0.835	0.840	0.845	0.874	
SEAa	0.895	0.898	0.899	0.900	0.902	0.909	
SEAg	0.892	0.896	0.896	0.897	0.899	0.906	
STAGGER	0.949	0.949	0.937	0.941	0.919	0.928	
LED	0.892	0.888	0.882	0.884	0.884	0.860	
MIXED_BALANCED	0.928	0.934	0.935	0.939	0.943	0.964	
MIXED_IMBALANCED	0.930	0.935	0.936	0.940	0.943	0.966	
RBF	0.850	0.868	0.873	0.875	0.878	0.881	
RBF_DRIFT	0.528	0.550	0.562	0.570	0.576	0.618	
WAVEFORM	0.814	0.822	0.826	0.828	0.832	0.834	
WAVEFORM_DRIFT	0.814	0.822	0.824	0.830	0.832	0.834	

Besides setting the chunk size, we fine-tuned the model hyper-parameters. Our main objective was to estimate the suitable combinations of the parameters α, β, and θ. As the experiments were computationally intensive, most of the models were trained using the default hyper-parameter settings, with particular values set to α = 0.002, β = 3, and θ = 0.02. Regarding the model behavior with different hyper-parameter settings, α influences the lifetime of an expert in the ensemble and the speed of degradation of the expert score with increasing lifetime. Increasing values led usually to a more dynamic model, able to adapt rapidly, while lower values led to a more stable composition of the model. β influences the preference of the experts, which classified the samples correctly. Higher values of this parameter can suppress the poor-performing experts and raise the probability of updating them in the following iteration. θ serves as a threshold for the expert update. Lower values usually lead to more weak experts in the ensemble, a marginal contribution to the performance , but raise the model complexity significantly. Higher values force the updating of weak experts. However, some of them may be missing later on, after drift occurrence and the reappearance of previous concepts.

The results proved, that the chunk size does have an impact on the model performance, and there are mostly minor differences in the performance metrics with different chunk size parameter setting. Although accuracy is not affected much, F1 metric improves significantly with larger chunk sizes, especially on the BNG_Zoo and BNG_Lymph datasets. In general, we can observe, that on the larger data (with more than 100,000 samples in the stream), larger windows resulted in slightly better performance. On the other hand, smaller chunk sizes enable the model to react more quickly to concept drift. In some cases, the accuracy metric proved to be not very useful, as the target class is strongly unbalanced or multi-class. It is evident mostly on the KDD99 or BNG_Lymph datasets, where high accuracy values are caused mainly by the classification into the majority class, while other minor classes do not influence this metric very much. A much better perspective on the actual model performance could be given by F1 measure.

The experiments summarized averaged results that the models achieved on the entire stream, but it is also important to explore how the performance progressed during the stream processing and observe their reactions to concept drift.

Figure 4 visualizes the accuracy achieved by the DDCW models on the real datasets with both chunk sizes. The performance of the method with both settings is overall similar; however, we can see a difference in cases when a change (possible drift) occurs. On the KDD99 dataset, there is a significant decrease in the accuracy of the model after around 52,000 samples. Shorter chunk windows resulted in a much earlier reaction to the change, without any significant decrease in performance. On the Elec and Covtype datasets, the earlier reactions are also present and visible, resulting in higher performance metrics.

Figure 4 Performance of the DDCW model on the real datasets.

(A) ELEC, (B) KDD 99, (C) airlines, (D) covtype, (E) shuttle, (F) powersupply, (G) connect-4, (H) BNG Bridges, (I) BNG Bridges1vsAll, (J) BNG Hepatitis, (K) BNG Zoo, (L) BNG Lymph (y axis) accuracy (x axis) number of samples.

Figure 5 depicts the DDCW model performance on the synthetic datasets with both chunk sizes. In the case of Stagger and LED datasets, the effects of different chunk sizes are comparable with the impact on the real datasets. Larger chunk sizes lead to later reactions to drift and a more significant decrease in accuracy. In contrast, performance evaluation on larger streams, such as AGR or Mixed streams, showed that the chunk size effect on stream processing is different. Contrary to previous datasets, larger chunk sizes resulted in a more robust model, still covering some of the previous concepts after drift occurrence. After the drift, the model performance dropped significantly. When using larger chunk sizes, the model was able to use more data to update the ensemble, which led to improved performance.

Figure 5 Performance of the DDCW model on the synthetic datasets.

(A) AGR_a, (B) AGR_g, (C) SEA_a, (D) SEA_g, (E) stagged, (F) LED, (G) mixed-balanced, (H) mixed-imbalanced, (I) RBF, (J) RBF Drift, (K) waveform, (L) waveform drift (y axis) accuracy (x axis) number of samples.

Figure 6 depicts the size of the ensemble during stream processing on selected datasets with the diversity of the ensemble members disabled and enabled. During the experiments, we set the minimum size of the DDCW ensemble to 5 and the maximum size to 20. Initially, we performed a set of tests to estimate the optimal ensemble size. Larger ensembles did not perform significantly better, while the computational complexity of the model was considerably higher. The experiments proved that, during the run time, the algorithm preferred a smaller pool of ensemble members. The algorithm added more ensemble members when a concept drift occurred. Enabling the diversity-based selection of experts resulted in a more stable composition of the ensemble and required fewer member updates.

Figure 6 Size of the DDCW ensemble during the run-time on the selected datasets.

(A) Airlines, (B) covtype, (C) stagger, (D) AGR_a (y axis) Number of ensemble members (x axis) number of samples.

Comparison with other ensemble models

In these experiments, we compared the DDCW model performance with the selected ensemble models. In the comparison, we included AWE, DWM, Online Boosting, OzaBagging, ARF, and KUE models. Each of the ensemble models was tested with different base learners. We evaluated Naive Bayes, Hoeffding Tree, and k-NN as base learners. Similar to the previous set of experiments, we used the accuracy, precision, recall, and F1 metrics for comparison purposes computed in the prequential fashion. When using the DDCW model, we included the DDCW model with a combination of Naive Bayes and Hoeffding trees as base learners, as well as a tree-based homogeneous ensemble alone.

Performance comparison

To summarize the experiments, Table 4 compares the performance of all evaluated models on the real datasets and Table 5 provides the similar comparison on the synthetic data streams. As in the previous experiments, the tables consists of overall averaged results that the models achieved on the entire stream. While most of the studies focus only on a comparison of accuracy, we decided to analyze other classification metrics as well. Especially in the case of multi-class or heavily imbalanced data (e.g., KDD99), accuracy might not be the best choice to truly evaluate the performance of the model, therefore we choose also F1 metric as well. Please note, that we were unable to properly obtain F1 values from the KUE model on some of the datasets.

Table 4 Comparison of accuracy and F1 metrics of evaluated ensemble models on the real data streams.

	DDCWHT	DDCWHTNB	DWMNB	AWENB	DWMHT	AWEHT	OBkNN	OzakNN	OBHT	OzaHT	OBNB	OzaNB	ARFHT	KUEHT	
Accuracy	
ELEC	0.853	0.810	0.800	0.756	0.869	0.788	0.765	0.780	0.858	0.793	0.792	0.734	0.857	0.668	
KDD99	0.995	0.991	0.983	0.420	0.989	0.103	0.999	0.998	0.998	0.995	0.995	0.946	0.999	0.999	
AIRL	0.662	0.649	0.640	0.618	0.620	0.575	0.587	0.639	0.634	0.653	0.619	0.644	0.666	0.663	
COVT	0.853	0.810	0.823	0.592	0.812	0.215	0.927	0.918	0.876	0.871	0.783	0.871	0.941	0.904	
SHUTTLE	0.996	0.992	0.896	0.949	0.946	0.949	0.990	0.991	0.982	0.978	0.950	0.922	0.998	0.997	
POWERSUPLY	0.158	0.158	0.074	0.186	0.074	0.187	0.029	0.167	0.013	0.162	0.007	0.161	0.156	0.218	
CONNECT4	0.719	0.705	0.671	0.286	0.693	0.613	0.702	0.728	0.675	0.696	0.636	0.572	0.739	0.650	
BNG_BRIDGES	0.737	0.725	0.611	0.698	0.612	0.698	0.621	0.670	0.699	0.750	0.687	0.684	0.756	0.738	
BNG_BRIDGES1vsAll	0.970	0.966	0.962	0.967	0.962	0.967	0.936	0.958	0.970	0.973	0.957	0.962	0.973	0.958	
BNG_HEPATITIS	0.909	0.897	0.854	0.877	0.856	0.913	0.839	0.884	0.913	0.920	0.868	0.853	0.922	0.923	
BNG_ZOO	0.928	0.921	0.812	0.913	0.806	0.913	0.909	0.927	0.931	0.915	0.903	0.889	0.942	0.939	
BNG_LYMPH	0.883	0.878	0.801	0.825	0.802	0.825	0.787	0.828	0.846	0.809	0.818	0.806	0.871	0.903	
F1	
ELEC	0.826	0.760	0.749	0.694	0.852	0.757	0.721	0.730	0.828	0.744	0.722	0.595	0.825	0.668	
KDD99	0.587	0.549	0.502	0.045	0.397	0.029	0.712	0.642	0.593	0.645	0.577	0.530	0.649	NaN	
AIRL	0.556	0.535	0.320	0.293	0.534	0.425	0.436	0.430	0.563	0.522	0.534	0.259	0.577	NaN	
COVT	0.826	0.760	0.594	0.156	0.569	0.085	0.740	0.718	0.721	0.675	0.626	0.679	0.779	NaN	
SHUTTLE	0.611	0.702	0.397	0.398	0.446	0.399	0.517	0.435	0.623	0.452	0.530	0.621	0.677	NaN	
POWERSUPLY	0.113	0.110	0.066	0.134	0.066	0.134	0.034	0.161	0.106	0.014	0.009	0.107	0.149	NaN	
CONNECT4	0.470	0.464	0.466	0.272	0.437	0.382	0.587	0.517	0.431	0.406	0.420	0.381	0.496	0.506	
BNG_BRIDGES	0.649	0.635	0.515	0.601	0.515	0.601	0.527	0.567	0.611	0.669	0.599	0.596	0.673	0.656	
BNG_BRIDGES1vsAll	0.895	0.883	0.871	0.886	0.869	0.886	0.790	0.857	0.899	0.906	0.852	0.872	0.906	0.915	
BNG_HEPATITIS	0.943	0.935	0.904	0.921	0.907	0.922	0.898	0.929	0.945	0.950	0.914	0.902	0.952	0.887	
BNG_ZOO	0.860	0.853	0.714	0.832	0.704	0.832	0.824	0.855	0.865	0.844	0.821	0.821	0.887	0.882	
BNG_LYMPH	0.740	0.752	0.610	0.603	0.609	0.603	0.602	0.584	0.735	0.658	0.679	0.659	0.636	0.827	

Table 5 Comparison of accuracy and F1 metrics of evaluated ensemble models on the synthetic data streams.

	DDCWHT	DDCWHTNB	DWMNB	AWENB	DWMHT	AWEHT	OBkNN	OzakNN	OBHT	OzaHT	OBNB	OzaNB	ARFHT	KUEHT	
Accuracy	
AGRa	0.918	0.879	0.770	0.821	0.810	0.822	0.611	0.662	0.818	0.913	0.816	0.658	0.926	0.943	
AGRg	0.896	0.874	0.750	0.800	0.792	0.800	0.600	0.649	0.794	0.845	0.786	0.658	0.897	0.943	
SEAa	0.890	0.888	0.876	0.879	0.874	0.878	0.753	0.863	0.871	0.868	0.846	0.854	0.894	0.902	
SEAg	0.887	0.884	0.874	0.876	0.872	0.875	0.750	0.860	0.867	0.868	0.845	0.854	0.891	0.910	
STGR	0.937	0.923	0.901	0.948	0.947	0.947	0.929	0.945	0.941	0.872	0.924	0.602	0.949	0.962	
LED	0.873	0.860	0.831	0.893	0.831	0.893	0.682	0.770	0.768	0.848	0.770	0.698	0.892	0.893	
MIXED_BALANCED	0.984	0.964	0.916	0.919	0.973	0.919	0.967	0.976	0.922	0.988	0.912	0.920	0.995	0.998	
MIXED_IMBALANCED	0.985	0.964	0.916	0.920	0.922	0.920	0.968	0.976	0.926	0.988	0.909	0.920	0.995	0.998	
RBF	0.924	0.882	0.712	0.733	0.712	0.733	0.915	0.940	0.816	0.940	0.719	0.720	0.941	0.954	
RBF_DRIFT	0.618	0.601	0.520	0.516	0.516	0.512	0.554	0.572	0.580	0.705	0.544	0.546	0.655	0.788	
WAVEFORM	0.840	0.837	0.797	0.830	0.797	0.830	0.762	0.815	0.844	0.855	0.807	0.805	0.843	0.844	
WAVEFORM_DRIFT	0.840	0.836	0.797	0.830	0.797	0.830	0.763	0.815	0.843	0.855	0.807	0.805	0.843	0.852	
F1	
AGRa	0.912	0.869	0.750	0.812	0.810	0.812	0.592	0.616	0.808	0.802	0.773	0.542	0.922	0.944	
AGRg	0.888	0.864	0.728	0.787	0.788	0.786	0.580	0.602	0.782	0.835	0.773	0.543	0.890	0.939	
SEAa	0.910	0.909	0.900	0.901	0.898	0.901	0.791	0.889	0.895	0.894	0.875	0.885	0.914	0.909	
SEAg	0.908	0.906	0.898	0.899	0.896	0.899	0.799	0.886	0.892	0.893	0.874	0.885	0.911	0.905	
STGR	0.941	0.928	0.947	0.951	0.949	0.949	0.934	0.947	0.945	0.887	0.930	0.633	0.952	0.963	
LED	0.873	0.860	0.832	0.893	0.832	0.893	0.682	0.768	0.768	0.848	0.770	0.693	0.892	0.893	
MIXED_BALANCED	0.984	0.964	0.917	0.919	0.973	0.919	0.967	0.976	0.923	0.988	0.912	0.920	0.995	0.998	
MIXED_IMBALANCED	0.985	0.966	0.923	0.925	0.928	0.926	0.969	0.977	0.928	0.989	0.914	0.927	0.996	0.998	
RBF	0.923	0.881	0.716	0.738	0.716	0.738	0.916	0.941	0.816	0.939	0.721	0.720	0.941	0.954	
RBF_DRIFT	0.620	0.619	0.545	0.533	0.521	0.517	0.559	0.585	0.578	0.703	0.577	0.595	0.644	0.788	
WAVEFORM	0.839	0.834	0.784	0.824	0.784	0.824	0.762	0.814	0.843	0.854	0.796	0.792	0.842	0.844	
WAVEFORM_DRIFT	0.840	0.834	0.784	0.824	0.784	0.824	0.763	0.814	0.842	0.854	0.795	0.792	0.842	0.852	

The DDCW model proved to be suitable for data streams with different concept drifts and either binary or multi-class classification tasks. When considering the composition of the DDCW ensemble, the fact that the model relies on different base learners enables it to utilize the strengths of particular learners. Dynamic composition of the ensemble enables it to adapt to the particular stream by preferring a base learner that is more suitable for the given data. In general, the DDCW performs very well on the generated streams, gaining at least competitive results compared to the other models on the real-world datasets. The method appears to struggle more with some of the imbalanced datasets, as is apparent from the F1 results achieved on the KDD 99 or Airlines dataset. During this experiment, we also used two different DDCW method setups to compare the effect of the base learner selection. We used DDCW with only Hoeffding trees as a base learner and DDCW with a combination of Naive Bayes and Hoeffding trees. Although the homogeneous ensemble mostly performed slightly better, the heterogeneous one was usually faster to train and score and maintained a similar performance, which was a result of the inclusion of fast Naive Bayes ensemble members. In a similar fashion, we experimented with an integration of k-NN into the DDCW model, but as expected, k-NN base learners raised the resource requirements of the model and failed to provide a sufficient performance boost, so we did not include k-NN base learners in further experiments.

Performance comparison showed that the DDCW method can produce results that are comparable to the related ensemble models on both, real and synthetic streams. In many cases, it is able to outperform existing algorithms (e.g., AWE, DWM, OZA, and OB) in both of the explored metrics. Current state-of-the-art methods such as KUE and ARF usually produce slightly better results, but the DDCW method showed a fairly competitive performance, surpassing on several datasets one of those methods in both, accuracy and F1 scores. However, the evaluation metrics represent only one aspect of the adaptive models’ performance. During the experiments, we tried to evaluate another aspect of the studied models that may influence the run time of the models during deployment in real-world scenarios. We focused mostly on monitoring the model performance in terms of their demand on resources and resource consumption during the process. During the experiments, we collected data about the overall run-time aspects of the model. The following section compares the models from the perspective of training/scoring times and memory requirements.

Training time and memory usage

We analyzed the training and scoring times and the memory consumption during the training process to provide a different view of the models’ performance, comparing performance metrics with resource consumption requirements. We measured the overall training and scoring times on the entire data by summing up all partial re-training and scoring times over the stream. Table 6 summarizes the results of all evaluated models. The table compares the total training time consumed in the training and re-training of the models, the total scoring time of all processed instances, and the average size of the model in memory during the entire stream processing. The results represent an averaged value of the total of five separate runs of each experiment. It is important to note that the KUE model was not included in this comparison. We used Python implementations of all evaluated models and the scikit-multiflow library during the experiments. The KUE model was available only in its Java implementation using the MOA library, therefore using different underlying technologies could influence the objective comparison of resource consumption. At the same time, it is essential to note that the Java implementation of the KUE model was significantly effective than all Python-based models, mostly in training times, which were remarkably shorter.

Table 6 Comparison of average total training and scoring times (in seconds) and average model size in memory (in kB).

	DDCWHT	DDCWHTNB	DWMNB	AWENB	DWMHT	AWEHT	OBkNN	OzakNN	OBHT	OzaHT	OBNB	OzaNB	ARFHT	
Train	
ELEC	143.26	111.69	45.11	129.48	13.57	20.79	1604.23	509.71	1,147.40	84.04	1,039.32	15.83	315.83	
KDD99	2,101.06	2,558.51	669.26	678.63	2,522.47	3,040.64	10,423.15	10,829.81	23,207.21	2,062.85	15,924.35	1,650.80	3,107.94	
AIRL	2,742.45	1,958.96	103.02	303.11	966.34	1,462.39	3,278.04	1,214.47	4,808.00	2,728.98	3,626.52	132.82	9,427.38	
COVT	143.26	111.69	486.37	4,316.73	8,679.77	19,836.23	6,433.28	5,748.82	52,179.02	15,154.78	25,203.72	9,532.05	8,798.81	
SHUTTLE	122.82	139.38	133.04	431.61	211.05	644.96	1,733.08	914.14	3,279.41	223.35	2931.21	45.55	582.35	
POWERSUPLY	554.17	452.09	70.71	305.48	225.03	757.13	2,984.78	644.79	19,965.02	408.03	10,153.15	28.47	773.20	
CONNECT4	887.58	810.98	306.71	1,189.89	486.08	1,189.16	7,480.17	3,040.63	6,988.81	678.24	9,876.03	184.61	1,267.93	
BNG_BRIDGES	20,336.45	15,750.75	2,623.52	11,661.23	7,238.69	23,103.62	105,021.95	42,943.55	23,113.54	6,226.40	19,416.33	987.16	18,992.29	
BNG_BRIDGES1vsAll	4,244.05	3,223.40	1,388.32	4,423.00	3,459.20	11,226.27	95,129.22	42,928.10	13,133.03	3,429.59	6,847.12	766.14	12,352.68	
BNG_HEPATITIS	7,234.04	5,903.23	1,909.77	6,160.89	5,921.37	17,505.05	142,511.73	61,844.19	17,275.70	5,807.02	12,077.91	1,333.14	22,191.77	
BNG_ZOO	8,833.11	7,587.96	831.53	3,786.11	3,085.02	9,840.18	23,267.58	11,843.88	14,623.41	2,335.37	17,983.05	983.95	24,609.51	
BNG_LYMPH	5,2199	60,795	27.21	95.20	121.78	312.60	1,666.61	633.64	7,795.36	147.28	13,468.46	1,049.58	16,545.83	
AGRa	1,836.59	1,729.20	1,255.87	4,079.91	1,909.92	6,579.19	33,991.85	14,658.03	7,781.19	1,774.19	5,960.05	627.15	13,707.31	
AGRg	2,867.83	2,731.58	1,228.79	3,910.93	1,350.43	4,336.84	33,765.55	14,634.41	7,825.25	2,044.13	5,961.76	627.01	14,255.32	
SEAa	1,390.82	1,144.45	537.41	1,355.54	903.92	2,407.77	25,996.58	11,367.54	4,050.33	945.55	2,600.97	243.14	14,138.98	
SEAg	1,938.19	1,562.57	535.09	1,358.02	898.35	2,375.66	25,845.26	11,367.24	4,052.31	924.03	2,535.35	237.78	10,720.81	
STGR	99.96	95.56	119.05	189.38	191.96	339.70	3,181.51	1,129.84	2,262.76	205.48	1,778.44	86.18	738.52	
LED	1,710.85	1,164.57	1,539	3,415	3,622.99	5,589.79	7,682	3,097	34,339.45	3,447.84	28,733.92	478.55	6,099.67	
MIXED_BALANCED	2,580.27	1,531.19	970.63	2,488.19	946.31	7,679.21	22,126.14	15,055.02	9,481.48	2,000.08	6,858.02	437.57	8,358.76	
MIXED_IMBALANCED	2,712.32	1,540.93	982.09	2,551.43	2,733.50	7,749.73	22,211.42	14,447.70	7,448.70	1,907.75	6,907.52	651.22	8,540.42	
RBF	8,090.26	5,870.58	1,517.56	4,729.81	3,503.11	10,058.28	18,172.70	16,260.66	13,485.50	3,528.25	9,509.59	986.65	19,776.25	
RBF_DRIFT	14,255.57	7,468.35	1,467.39	4,411.77	2,433.44	6,011.22	38,684.40	16,204.23	89,52.23	2,793.67	9,863.56	955.75	20,931.48	
WAVEFORM	18,050.07	10,330.24	3,292.84	16,965.00	10,035.01	26,579.84	59,396.41	25,562.21	25,414.68	10,446.38	17,079.50	1,394.92	37,111.64	
WAVEFORM_DRIFT	19,060.56	9,819.87	3,217.57	12,837.63	5,691.92	17,527.29	59,255.53	25,508.46	25,871.33	10,373.11	16,140.68	1,368.99	36,135.95	
Scoring	
ELEC	44.18	44.59	30.08	60.34	8.76	5.92	383.55	399.24	46.97	54.61	39.86	35.49	30.33	
KDD99	826.72	1,789.61	582.33	261.48	1,108.80	938.46	5,220.47	10,181.99	3,072.03	1,653.69	6,070.47	29,248.61	454.54	
AIRL	682.19	642.45	69.00	137.29	296.88	659.61	964.92	972.81	882.57	1,436.29	765.43	175.72	709.79	
COVT	44.18	44.59	348.98	952.85	3,345.07	6,734.88	2,674.34	5,429.85	11,320	7,799.45	8,935.04	4,950.78	1,190.09	
SHUTTLE	43.83	74.66	106.36	190.00	101.52	237.67	546.34	756.35	200.45	188.33	254.86	263.51	83.27	
POWERSUPLY	206.63		186.37	124.85	143.21	263.09	547.29	468.14	678.93	680.06	398.07	298.00	58.48	
CONNECT4	119.21	180.59	225.28	448.38	119.71	264.27	2,555.47	2,831.78	452.03	204.62	621.94	633.43	125.87	
BNG_BRIDGES	3,658.96	4,447.50	2,192.79	4,485.01	3,452.78	7,233.22	40,374.27	40,626.88	6,059.87	3,820.15	6,090.57	5,606.51	1,623.26	
BNG_BRIDGES1vsAll	1,228.14	1,337.69	932.74	1,872.88	1,336.98	3,714.29	3,985.74	40,591.08	2,963.44	2,264.16	1,954.08	1,759.19	1,253.93	
BNG_HEPATITIS	1,432.58	1,755.10	1,289.06	2,554.61	2,349.87	5,490.33	58,901.34	59,433.80	4,161.07	2,695.03	3,287.22	3,064.09	1,353.66	
BNG_ZOO	1,655.25	2,796.35	711.68	1,419.12	1,406.56	3,003.11	10,138.36	11,355.23	1,877.31	1,299.39	7,308.81	6,687.30	294.30	
BNG_LYMPH	2,265.33	3,324.49	20.80	40.57	52.41	104.56	486.95	584.83	92.46	101.46	4,520.05	4,209.24	12.97	
AGRa	589.06	681.66	859.27	1,695.15	825.03	2,156.90	12,148.39	12,004.23	1,676.38	1,247.63	1,569.22	1,465.04	1,375.23	
AGRg	842.44	1,030.88	842.34	165.97	557.85	1,466.99	11,938.29	12,016.64	1,684.98	1,328.85	1,576.46	1,516.61	1,375.41	
SEAa	491.00	463.57	366.80	721.22	475.68	929.90	9,368.47	9,501.99	899.43	753.90	616.91	565.17	980.09	
SEAg	690.24	669.94	363.96	717.64	471.14	927.79	9,309.85	9,463.46	871.64	735.32	602.78	553.37	802.78	
STGR	43.36	46.87	70.61	144.67	86.07	166.89	935.82	891.94	173.56	155.34	140.56	122.58	96.58	
LED	544.33	603.01	1,305.11	2,665.37	1,610.95	3,274.27	2,777.76	2,834.50	3,019.31	2,594.30	2,753.80	2,781.52	528.32	
MIXED_BALANCED	1,118.60	737.17	655.21	1,280.62	533.98	2,711.89	8,627.57	12,252.98	2,219.60	1,939.11	1,701.65	940.76	1,094.69	
MIXED_IMBALANCED	1,189.05	748.56	667.03	1,308.05	1,459.17	2,723.83	8,646.08	11,612.31	2,214.76	175.07	1,746.56	1,432.22	1,119.65	
RBF	1,778.62	2,104.27	1,053.82	2,084.06	1,626.98	3,302.48	6,634.78	13,305.02	2,861.86	2,219.82	2,500.18	2,238.74	1,765.03	
RBF_DRIFT	2,592.40	1,755.61	1,013.02	1,981.98	1,028.25	2,006.14	13,536.52	13,383.73	2,039.09	1,956.97	2,457.97	2,199.96	2,596.57	
WAVEFORM	2,738.06	2,474.19	2,501.26	4,900.75	4,298.73	7,585.06	22,613.52	22,503.25	5,233.95	5,764.22	4,922.47	4,636.79	3,266.90	
WAVEFORM_DRIFT	2,934.28	2,234.43	2,460.86	4,947.01	1,028.25	2,006.14	22,610.44	22,508.14	2,039.09	1,956.97	4,672.90	4,375.81	3,250.45	
Memory	
ELEC	4,509	3,166	52	383	169	186	4,116	4,468	1,994	2,180	181	101	1,502	
KDD99	18,891	18,825	976	1,872	1,703	1,888	33,283	41,683	1,610	20,538	2,150	10,128	1,335	
AIRL	21,508	19,926	56	436	64	451	4,828	4,868	3,776	23,113	224	115	34,904	
COVT	4,509	3,166	489	5,419	863	5,432	6,821	23,218	3,355	201,265	1,838	195,008	9,484	
SHUTTLE	4,664	2,952	136	779	371	797	3,986	5,640	310	1,114	412	330	3,376	
POWERSUPLY	2,653	2,664	129	899	133	914	7,478	7,610	981	642	926	627	21,425	
CONNECT4	53,333	46,010	372	2,724	376	2,742	18,579	18,527	1,465	12,246	878	782	6,267	
BNG_BRIDGES	42,056	37,951	201	1,375	206	1,392	6,970	6,877	2,119	27,417	561	399	101,039	
BNG_BRIDGES1vsAll	43,681	44,147	98	713	430	730	6,941	6,855	156,252	53,302	308	192	279,918	
BNG_HEPATITIS	47,294	47,956	150	1,084	155	1,102	10,010	9,857	102,517	108,163	401	296	234,600	
BNG_ZOO	38,211	40,955	308	2,079	313	2,097	8,550	8,800	21,231	9,192	718	614	24,674	
BNG_LYMPH	52,199	60,795	199	1,384	204	1,401	9,188	9,134	1,092	408	556	401	12,007	
AGRa	37,091	35,128	74	535	133	562	5,739	5,647	632	37,433	278	145	40,581	
AGRg	33,633	31,176	74	545	75	562	5,726	5,647	594	56,528	288	145	29,348	
SEAa	33,881	33,449	29	220	34	239	3,435	3,341	11,497	21,888	157	55	206,872	
SEAg	32,350	31,976	28	220	33	238	3,430	3,340	9,279	22,056	156	55	121,890	
STGR	3,794	3,661	29	220	59	138	3,391	3,318	599	627	163	55	479	
LED	4,712	2,235	553	3,680	558	3,694	11,580	11,505	1,068	9,890	1,182	1,103	2674	
MIXED_BALANCED	34,658	34,182	36	276	789	292	2,767	3,794	204	9,199	180	69	15,355	
MIXED_IMBALANCED	35,603	34,243	37	276	231	293	1,738	3,791	410	9,369	179	69	16,163	
RBF	45,067	38,146	82	598	86	616	2,835	6,323	572	65,277	267	160	169,534	
RBF_DRIFT	44,461	43,113	82	598	86	615	6,222	6,130	1,283	59,310	311	160	45,777	
WAVEFORM	54,302	49,396	200	1,419	204	1,432	10,797	10,639	49,221	68,083	557	395	371,158	
WAVEFORM_DRIFT	56,276	43,703	200	1,419	204	1,433	10,799	10,640	53,793	69,302	557	395	380,476	

The choice of a base classifier heavily influences the overall run-time requirements of all models. Most apparently, the choice of k-NNADWIN as a base learner for OnlineBagging and OzaBagging methods leads to a massive increase of memory consumption and data processing times. Nearest neighbor classifiers require to store the training data in memory, which could lead to increased memory consumption and therefore increased training times. On the other hand, ensembles which consist of Naive Bayes and Hoeffding Tree classifiers as the base learners are much faster to train and require significantly lower memory during the run-time. However, Online Boosting and OzaBagging methods are much more sensitive to the number of features of the processed data. It can be observed on KDD99 and Covtype datasets, where these relatively faster models, required significantly longer times to either train or score the instances. DDCW ensemble training time and memory consumption requirements reflect the fact that the model consists of a mixture of Hoeffding Tree and Naive Bayes classifiers. When experimenting with the homogeneous DDCW ensemble, the performance results were better on many of the datasets. On the other hand, heterogeneous DDCW model provided a small decrease of the performance, but in most of the cases, inclusion of a Naive Bayes base learner led to a shorter training times and reduced memory usage (most significantly on e.g., Waveform data, where the training time was reduced to a half of the total training time of the homogeneous DDCW ensemble). When taking into consideration both aspects, DDCW model can, in some cases present a compromise between performance and resource requirements. Although Online Boosting or OzaBagging model performs with higher degrees of accuracy on some of the datasets, their computational intensiveness and more extended training and scoring times may be a factor to consider a simpler model. Similarly, ARF and KUE models provide superior performance on the majority of the datasets. When compared to those state of the art methods, DDCW method produced mostly comparable results, but needed less training time with lesser memory requirements (especially on the larger synthetic data streams) than ARF method. DDCW ensemble, in this case, may offer a reasonable alternative by providing a well-performing model, while maintaining reasonable requirements on run-time.

Conclusions

In the presented paper, we propose a heterogeneous adaptive ensemble classifier with a dynamic weighting scheme based on the diversity of its base classifiers. The algorithm was evaluated on a wide range of datasets, including real and synthetic ones, with different types of concept drift. During the experiments, we compared the proposed method with several other adaptive ensemble methods. The results proved that the proposed model is able to adapt to drift occurrence relatively fast and is able to achieve at least comparable performance to the existing approaches, on both real and synthetically generated datasets. While still performing well, the model also manages to maintain reasonable resource requirements in terms of memory consumption and time needed to score the unknown samples. The proposed approach is also dependent on chunk size parameter setting, as the performance of the model on certain datasets change significantly with different chunk sizes. Further research with adaptive heterogeneous ensemble models may lead to an exploration of modifications to weighting schemes that improve performance in multi-class classification problems or classifications of heavy imbalanced data. Another interesting field for future work is the integration of adaptation mechanisms with semantic models of the application domain. A domain knowledge model could provide a description of the data, the essential domain concepts, and their relationships. Such a model could also be used to improve classification performance by capturing expert domain knowledge and utilizing it in the process of classification of unknown samples. A knowledge model could be used to extract new expert features not previously contained in the data or to extract interesting trends present in the data stream. Such extensions could represent expert knowledge and could thus be leveraged to detect frequent patterns leading to concept drift while reducing the time normally needed to adapt the models with that knowledge.

Additional Information and Declarations

Competing Interests

Author Contributions

Data Availability

The authors declare that they have no competing interests.

Martin Sarnovsky conceived and designed the experiments, performed the experiments, analyzed the data, prepared figures and/or tables, authored or reviewed drafts of the paper, and approved the final draft.

Michal Kolarik conceived and designed the experiments, performed the experiments, analyzed the data, performed the computation work, prepared figures and/or tables, authored or reviewed drafts of the paper, and approved the final draft.

The following information was supplied regarding data availability:

Source codes of the model is available at GitHub: https://github.com/Miso-K/DDCW.

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
