# Peer review of "Classification of the drifting data streams using heterogeneous diversified dynamic class-weighted ensemble"

_PeerJ Computer Science, doi:10.7717/peerj-cs.459_

## Round 0.1 · original submission · Major Revisions

Three reviewers have provided meaningful comments to improve the manuscript. Please review and resubmit.

·

Basic reporting

Authors propose a new ensemble method for learning from drifting data streams, which is always a commendable and interesting research direction. The paper certainly is well-written and the idea of using a heterogeneous ensemble is always interesting, as most works in data stream mining focus on homogeneous ensembles. The idea of the proposed ensemble is clearly presented and easy to follow. However, there is a couple of things that must be improved regarding the presentation, positioning the proposed ensemble among state-of-the-art approaches, as well as the experimental design. Regarding the manuscript properties falling under "basic reporting" part, my main comments are to be found below:

1. Please update your figures, make them more readable, scale them properly, as in many figures there is a lot of white spaces and plots are squeezed to a very narrow range
2. There is no need to describe used data sets, they are well-known.
3. Please update your literature on data stream mining, ensemble learning for data streams, role of ensemble diversity in data stream mining, and learning under limited supervision. Various recent papers published in 2018-2020 period are missing.

Experimental design

Experimental setup needs a major revision and improvement before this paper can be considered further. My main comments are below:

1. Used real data streams are old and very simple. There are more recent real data stream benchmarks published recently in the literature. Please extend your study with at least several of them.
2. Your experimental setup misses comparison with two state-of-the-art ensemble methods that are currently considered as best ensemble learners for drifting data streams - Adaptive Random Forest and Kappa Updated Ensemble. Please include comparison with those algorithms to make your experimental study relevant. At this point you compare yourself only with ensemble algorithms that are known to underperform when compared to ARF and KUE.
3. Do you calculate your performance metrics in a prequential fashion?
4. How does your method work under limited access to class labels? Assumption that you work with a fully labeled data stream is unrealistic. It would be interesting to add an experimental section, where your evaluate the impact of limited supervision on your method.

Validity of the findings

I will need to re-evaluate the conclusions and findings after Authors improve the experimental section. At this point, I find them well-written and interesting.

Additional comments

Well-written and interesting research paper, but requires a major update of the experimental section in order to make it comparable to the state-of-the-art.

Reviewer 2 ·

Basic reporting

The paper has several issues in writing and notation that should be addressed. Furthermore, there are important references missing. More details are provided in the general comments to the authors.

Experimental design

The analysis conducted has a decent protocol, yet, there are missing relevant works that should be used in the experimentation. Without proper assessment against state-of-the-art classifiers, it is impossible to determine if the results are conclusive.

Validity of the findings

Following the previous comments, it is impossible to determine whether the results are competitive against state-of-the-art ensemble classifiers. The manuscript, in its current state, does not allow a proper assessment.

Additional comments

C1: The manuscript lacks important references. For instance, in the introduction, authors bring forward a description for concept drifts, yet, no reference Is provided. Authors should add references for definitions that are not novel.

C2: Furthermore, I believe that some references are not entirely accurate. For instance, in the Background section, authors state the following: “The main idea of the ensemble model is based on the assumption that a set of classifiers together can achieve better performance than individual classifiers (Gama et al. (2008))”. Even though this reference is valid, there are seminal works like the one from Kuncheva that should be analyzed and acknowledged.
- Kuncheva L.I., C.J. Whitaker, C.A. Shipp and R.P.W. Duin. Is independence good for combining classifiers?, Proc. 15th International Conference on Pattern Recognition, Barcelona, Spain, 2000, 2, 168-171
- Kuncheva L.I., C.J. Whitaker. Measures of diversity in classifier ensembles and their relationship with the ensemble accuracy, Machine Learning, 51, 2003, 181-207


C3: Authors are also encouraged to conduct a thorough review of the text, as there are missing commas and also commas that should be removed. Some examples I extracted from the introduction but that occur in other parts of the manuscript:
- The primary motivation of this study is to design a model, capable of handling various types of drifts.
- We assume, that the creation of a heterogeneous ensemble consisting of different base learners with adaptation mechanism which ensures the diversity of its members can lead to a robust ensemble, able to handle the drifting data efficiently.
C4: Another important paper that should be acknowledged in terms of related works is HEFT-Stream. Despite the underlying goal is to overcome feature drifts, it is also an heterogeneous ensemble for data streams with concept drift detection.

C5: The notation used in the section that describes the proposed method should be revisited. For instance, authors use $E_1 – E_m$, which should be replaced with $E_1, \ldots, E_m$ as there is no subtraction involved. Another thing that should be clarified is the notation used in Figure 2: what does $X e \{1, \ldots, n\}$ stand for? Furthermore, there are sentences like “represented as a matrix $W m \times c$ which are readable but not formally correct. Authors are then encouraged to revisit the entire notation.

C6: Some questions and on algorithm 1: what is time? What is “treshold”? Shouldn’t it be threshold? Furthermore, is this the same as \theta?

C7: Authors must revisit the dataset descriptions provided in Table 2. Is the dataset said to be abrupt and gradual, or are the drifts abrupt and gradual? Furthermore, are the drifts real or synthetic? Please be consistent with what is depicted in the text and table label.

C8: I am highly interested in Figures 3 to 6. What exactly do they mean? How can we verify that the data distribution is changing? What do the colors and bullet sizes depict?

C9: There are relevant ensembles that are not being compared in the experiments section. Examples include ARF and KUE:
Gomes, H.M., Bifet, A., Read, J. et al. Adaptive random forests for evolving data stream classification. Mach Learn 106, 1469–1495 (2017). https://doi.org/10.1007/s10994-017-5642-8
Cano, A., Krawczyk, B. Kappa Updated Ensemble for drifting data stream mining. Mach Learn 109, 175–218 (2020). https://doi.org/10.1007/s10994-019-05840-z
Without a proper comparison with state-of-the-art ensembles, it is impossible to determine whether the results depicted in the paper are competitive.

Reviewer 3 ·

Basic reporting

The related works, presentation of the algorithm, style, structure of the article are acceptable. The language should be polished, because there are several errors, eg., in the abstract (the second sentence)
"Streams are often very dynamic, and its underlying" - their underlying ...
There are some parts of the article which are hard to understand. I suggest asking a native speaker to review the language or send the article to a proofreading company.
The presented figures are clear, but I suggest changing fig.2 - to be more understandable for readers. Some of the definitions need to be more precise - I think that author should define data stream, concept drift, decision rule, etc.

Experimental design

The goals of the experiment are unclear - what research question should be answered during this research?
The experimental protocol is unclear. How the authors count the metrics - I assume that test-then-train protocol was used (which is appropriate for the chunk-based approach), but then they should answer the question of how online learners have been modified because they used e.g., online bagging.
There is a lack of statistical analysis of the results.
The authors mentioned that the scope of the experiments was wider (e.g., they run experiments for different chunk size)- I understand that not all results should be included, but then a link to the repository or supplementary material should be added.
Nowadays most of the researches are replicable research, then access to the code is required to allow the readers to repeat the experiments.

Validity of the findings

Because the research questions were not formulated, therefore the conclusions are not so precise. I propose to add (after experimental research section) lessons learned section, where the authors may analyze the behavior of their method.
The idea is similar to the horse racing approach - a few words about this is needed.
I propose to run several experiments on a more controllable environment to check how the proposed algorithm reacts to the different types, severity, and frequency of drifts. I think it will behave pretty well for slow changes.
There is a lack of discussion on the weights accumulations - why it is so important, how this will impact the quality of the proposed solution.
Some important state-of-art methods should be added to comparison,e.g., from Learn++ family.

Additional comments

see above

---

## Round 0.2 · Minor Revisions

Please address the comments of the reviewers.

Reviewer 2 ·

Basic reporting

C1: The paper still requires proofreading. There are still some sentences that are somewhat unclear and some words that should be replaced. For instance, there are many places where “the” should be replaced either by “a” or “an”, and vice-versa.

C2: The description provided in lines 255 to 269 must be clarified. What is Q statistic? How is it computed? This might be common ground for researchers in the area, but authors cannot assume all readers are familiar with ensemble diversity quantification.

C3: Following my previous comment, line 265 is unclear as it mixes mathematical notation with textual descriptions to declare how weights are computed. Again, please be specific and clear on these notations.

C4: Line 10 of Algorithm 1 is unclear.

C5: Authors state in lines 369-371 that Gini index was used to compute feature importance. This is an interesting approach, yet, the computation of Gini index is not clarified. Please provide a mathematical formulation for this. This is relevant as Gini index is several times confused with Gini impurity.

C6: Figure 3 and how it should be read has been clarified in its caption (x and y axis, bullet sizes, etc), yet, this information should accompany lines 373 and 374.

C7: Regarding the y axis of Figure 3, should it be “Feature Indices” instead of “Features”?

C8: Please increase the font and legend sizes in Figures 4 to 6.

Experimental design

The experimental design has improved from the previous round. I am satisfied with the experimental protocol adopted expect for the comment below.

C1: In addition to providing the number of classes available in each dataset, please clarify the class ratio in each dataset. This is relevant as accuracy may not be appropriate for all datasets. For instance, KDD99 is imbalanced (without mentioning that it is highly biased according to [1], and thus, the assessment is flawed. It is important to make sure that the same does not happen with other datasets.

[1] M. Tavallaee, E. Bagheri, W. Lu, and A. Ghorbani, “A Detailed Analysis of the KDD CUP 99 Data Set,” Submitted to Second IEEE Symposium on Computational Intelligence for Security and Defense Applications (CISDA), 2009.

Validity of the findings

The claims are backed up by the results. Nonetheless, please refer to the comment above for class imbalance analyses.

Additional comments

In this revised version, the authors have addressed my previous review’s comments. Nonetheless, I still have a couple of concerns w.r.t. the presentation and analyses of the results that I have detailed in the specific comments.

Reviewer 3 ·

Basic reporting

no comments - this version could be accepted with minor improvements, as a discussion on weights are still missing

Experimental design

chosen metrics are disputable, but I accept them
It is ok, that you added a few reference methods, but Learn++ should be also added, especially because you deal with online learning.

Validity of the findings

I still cannot find the discussion on the weights accumulations - why it is so important, how this will impact the quality of the proposed solution. The authors did not answer this.

Additional comments

see above

---

## Round 0.3 · accepted · Accept

1) Please include average results for Accuracy and F1 for each of the methods in the tables, as a bottom row

2) Please revise the bibliography and complete the references of the papers (page numbers, volume, etc) that are often missing.

Congratulations.